# COVID-19 Vaccination Improved Psychological Distress (Anxiety and Depression Scores) in Chronic Kidney Disease Patients: A Prospective Study

**DOI:** 10.3390/vaccines10020299

**Published:** 2022-02-16

**Authors:** Helena Garcia-Llana, Nayara Panizo, Lorena Gandía, Marisa Orti, Elena Giménez-Civera, Claudia Forquet, Luis D’Marco, Maria Jesús Puchades, Mari Sargsyan, Irina Sanchís, Carmen Ribera, Mª Inés Marco, Cristela Moncho Ferrá, Carmen María Pérez-Baylach, Begoña Bonilla, Francesc Moncho Francés, Elisa Perez-Bernat, Asunción Sancho, Jose Luis Górriz

**Affiliations:** 1Hospital Universitario La Paz-IdiPAZ, Nephrology Department, Universidad Internacional de La Rioja (UNIR), Centro de Estudios Superiores Cardenal Cisneros, Universidad Pontificia de Comillas de Madrid, 28006 Madrid, Spain; helena.garcia@unir.net; 2Nephrology Department, Hospital Clínico Universitario de Valencia, INCLIVA Health Research Institute, 46010 Valencia, Spain; lorenagandiaincliva@gmail.com (L.G.); gimenez_eleciv@gva.es (E.G.-C.); claudiaforquet@gmail.com (C.F.); chuspuchades@gmail.com (M.J.P.); sargsyan_mari@gva.es (M.S.); sanchis_iri@gva.es (I.S.); dpca_hcv@gva.es (C.R.); marco_maradr@gva.es (M.I.M.); moncho_crifer@gva.es (C.M.F.); frances_fra@gva.es (F.M.F.); perez_eliber@gva.es (E.P.-B.); gorriz_jos@gva.es (J.L.G.); 3Preventive Medicine Department, Hospital Clínico Universitario de Valencia, 46010 Valencia, Spain; orti_raf@gva.es; 4Universidad Cardenal Herrera-CEU, CEU Universities, 46115 Valencia, Spain; luisgerardodg@hotmail.com; 5B-Braun AvitumValnefron Massamagrell Dialysis Centre, 46130 Massamagrell, Spain; carmen_maria.perez@bbraun.com; 6B-Braun AvitumValnefron Valencia Dialysis Centre, 46021 Valencia, Spain; begona.bonilla@bbraun.com; 7Nephrology Department, Hospital Universitario Dr. Peset, 46017 Valencia, Spain; sancho_asucal@gva.es; 8Department of Medicine, University of Valencia, 46010 Valencia, Spain

**Keywords:** COVID-19, vaccination, psychological distress, chronic kidney disease

## Abstract

The purpose of the study is to analyze the impact of vaccination against SARS-CoV-2 on anxiety and depression scores in patients with different modalities of chronic kidney disease. One hundred and seventeen renal patients (50 hemodialysis patients, 13 peritoneal dialysis patients, 32 kidney transplants, and 22 advanced chronic kidney disease patients at pre-dialysis care) were evaluated for depression, anxiety, health-related quality of life (HRQOL), and perceived fears and resources with standardized (Hospital Anxiety and Depression Scale (HADS)) and self-reported questionnaires. The measure points were before vaccination and 15 days after vaccination. The main finding of the study was that there was a decrease in the global mean of normal scores for anxiety and depression symptoms in chronic kidney disease patients post-vaccination. We did not find statistically significant differences in depression or anxiety scores, nor any HRQOL differences between the treatment groups. The three main fears reported by the participants at baseline were those of adverse effects, not getting the vaccine, and lack of information. These findings highlight the potential interest of assessing psychological variables related to the impact of vaccination against SARS-CoV-2. New studies will be required to assess the impact of comprehensive vaccine coverage and its psychological impact.

## 1. Introduction

The COVID-19 pandemic has caused a notable impact on the mortality and quality of life of billions of people all over the world, and those with chronic diseases such as chronic kidney disease have been especially affected. It is natural that protection of the physical integrity of the population has been the focus of efforts thus far. The pandemic, however, also represents a current and future threat to the mental health status of many individuals [1].

CKD patients have shown alterations in anxiety and depression scores [2]. Given that severe psychological distress was already known to be highly prevalent in those patients [3], the pandemic represented a potentially aggravating situation. It is for this reason that many authors have paid special attention to the impact of the COVID-19 pandemic on the psychological well-being of these patients [4]. 

Vaccination is one of the most effective ways to stop the spread of COVID-19 infection, but while safe vaccines have been developed, there has been a growing distrust of them in some sectors of the population [5]. Within this context, people living with chronic kidney disease may suffer confusion and psychological impact, as well as other psychological effects, as shown by previous research [6,7,8]. Moreover, high mortality rates in CKD patients have been recorded after SARS-CoV-2 infection, especially those patients with kidney transplants and on dialysis [9,10].

All these disconcerting factors may aggravate psychological distress in patients with CKD. To the best of our knowledge, the psychological impact of COVID-19 vaccination has not yet been evaluated for the general population nor for CKD patients. 

The objective of our study was to analyze the impact of vaccination against SARS-CoV-2 on anxiety and depression scores in patients with different modalities of chronic kidney disease.

## 2. Materials and Methods

### 2.1. Study Design and Settings 

#### 2.1.1. Participants

A non-random sample of one 117 renal patients from a nephrology department of two university hospitals and two hemodialysis clinics from the region of Valencia were assessed for the present study.

The assessment protocol was made up of four sections, as described below.

#### 2.1.2. Instruments

Assessments were administered prior to the first vaccine first and 14 days after the second vaccine dose. They included self-reported questionnaires and clinically-related measures. Cronbach’s alpha coefficients were calculated for the current sample (*n* = 117) and are presented in parentheses. The self-reported questionnaire consisted of the following parts.

Survey of the sociodemographic data and clinical profile. This survey collected data on age, gender, type of treatment (advanced chronic kidney disease not-on-treatment, hemodialysis, peritoneal dialysis or transplantation), years on treatment, and other comorbidities. 

The Hospital Anxiety and Depression Scale (HADS) [11] is a 14-item questionnaire with a Likert-type response format of four points (0–3); seven questions correspond to the anxiety subscale (HADS-A) and the other seven correspond to the depression subscale (HADS-D). Higher scores indicate greater severity. The cut-off scores for the subscales are as follows: 0–7 normal, 8–10 mild/doubtful, and >10 moderate-severe (clinical anxiety or depression). A value greater than 10 suggests the existence of a clinical problem of anxiety or depression, or both, according to the score in each of the two subscales.

The HADS has been shown to have adequate psychometric properties in various groups of patients, including patients with kidney disease treated with hemodialysis [12]. The Spanish version [13] has shown adequate internal consistency (α = 0.86) and concurrent validity. In this study, the internal consistency index was α = 0.83 for the global scale and y = 0.79 and 0.61 for the subscales HADS-A and HADS-D, respectively. 

For the survey on fears and resources regarding vaccine, the third section explores the patient fears and coping resources through three open-ended questions.

“Regarding the vaccine, what do you fear most?”“Regarding vaccine-related fears, what do you think helps you feel better?”“How do you think we can help you from the hospital/hemodialysis clinic?”

Health-related quality of life (HRQOL) is the fourth section, which explores a patient’s health-related quality of life and is assessed by asking one open-ended question on a 11-point Likert scale, with scores ranging between 0 (very poor) and 10 (excellent).

“How would you describe your quality of life in general?”

This question has been shown to have adequate psychometric properties in patients with kidney disease treated with hemodialysis [14]. 

#### 2.1.3. Procedure 

Approval from the Clinical Research Ethics Committee of one of the hospitals involved was obtained before beginning the study (Hospital Clinico Universtario de Valencia, date of approval: 21 February 2021). The hospital-based nephrology department attends to 42 prevalent stable HD patients, 85 patients on peritoneal dialysis, and 60 patients with CKD not yet on a dialysis program. Hospital Universitario Doctor Peset Valencia attends 38 stable patients with a kidney transplantation received within the last 3 years. The two hemodialysis clinics attend 192 prevalent stable patients. From 22 February to 30 April 2021, all the patients who met the inclusion criteria (18 years or older, no diagnosed psychiatric disease, ability to understand the assessment protocol, and having signed the informed consent) were offered the possibility of participating in this study. One hundred and seventeen patients (28% of the total stable patients in the four settings of assessment) accepted and were assessed by a trained nephrologist when they came for their COVID-19 vaccine appointment and then 14 days afterwards.

### 2.2. Sample Size Calculation 

Using previous studies that have found significant differences in the scores of the HADS test on anxiety and depression in patients with chronic kidney disease [15], the standard deviation score for depression and anxiety was 3.8 and 4.0, respectively. Assuming a two-sided paired alpha error of 0.05, power of 80% correlation between pre-post treatment of 0.7, and drop-out rate of 20%, the total sample size required was 15 patients per group.

### 2.3. Data Analysis 

Descriptive statistics were calculated, including means, SDs, ranges for quantitative variables, and frequency tables for the qualitative variables. The psychological variables were assessed at two time points—before the first vaccine dose and 14 days after the second dose. Given the distribution, quantitative variables were expressed as median and interquartile range, and qualitative variables as percentages. Mann−Whitney U test was used to compare differences between continuous variables in the two independent groups. The Chi-square test was performed for non-dichotomous qualitative variables at the two assessment points. The ANOVA test was used for comparing the means of different samples in more than two groups.

For all analyses, two-tailed tests were used to determine statistical significance. The IBM SPSS Statistics version 21.0 software package 21.0 version was used. 

## 3. Results

### 3.1. Psychological Distress and HRQOL Pre and Post Vaccination

The HADS survey was filled out by 117 patients pre-vaccination (50 hemodialysis patients, 13 peritoneal dialysis patients, 32 kidney transplant patients, and 22 advanced chronic kidney disease patients on pre-dialysis care). Their mean age was 65 and the median was 68 (57–74) years (mean [interquartile range] (IQR)). Of the participants, 78 were male and 39 female (67 and 33%)—all of whom received complete vaccination. Details of the sociodemographic characteristics and clinical profile of the sample are summarized in Table 1. 

Only 87 participants answered the survey 15 days after vaccination; this is the population in which the analysis was carried out.

Figure 1a shows the scores of the study patients on the anxiety scale. Although most of the scores fell within normality on the anxiety subscale, we found that before vaccination 8% of patients had scores greater than 11, suggestive of a clinical problem. After vaccination, this percentage of people decreased to only 1%, while people with normal results increased to 7%. In patients with intermediate or doubtful results, there was hardly any change in anxiety scores after vaccination. That is, after vaccination, anxiety scores improved mainly in those patients in whom anxiety was a true clinical problem.

In the depression subscale, the results were similar; the percentage of people with scores in the normal range was also higher after vaccination. However, both before and after vaccination, most patients presented results within the normal range. 

Figure 1b shows similar results on the depression scale, but in this case the percentage of patients with results compatible with a clinical problem was very low, both before and after vaccination. Regarding the depression score, a slight decrease occurred in the doubtful scoring band after vaccination, while the percentage of patients with results considered within the normal range increased.

We were not able to assert that these results were directly related to vaccination because the study was not designed to confirm this hypothesis; there was no control group and we are aware that many other personal circumstances may act as a bias.

The comparison analysis of the qualitative proportions of the three cut-off scores for the subscales of the anxiety and depression tests did not show significant differences, although a certain trend was found in the case of anxiety (*p* = 0.05). The comparison of scores in the form of a continuous variable for both anxiety and depression, however, showed a significant improvement and benefit before and after vaccination (Figure 2). This result occurred despite the fact that the majority of patients started with scores within the normal range.

We did not find statistically significant differences in depression and anxiety scores between the different groups of treatment (HD, PD, KT, or ACKD) patients (ANOVA test, *p* = 0.10; Figure 3). There are some aspects of the study, in relation to this fact, that are worth highlighting. Firstly, the hemodialysis group maintained very similar scores before and after vaccination for both anxiety and depression, with hardly any changes after vaccination. It is striking, however, that patients who were not yet undergoing renal replacement therapy (advanced CKD) had higher scores for both anxiety and depression, both pre- and post-vaccination. 

No significant differences were found between the age groups in both the anxiety and depression scores in the HADS test (*p* = 0.07 and 0.38, respectively). It is noteworthy that the highest anxiety scores were presented by patients who were in the youngest quartile. In addition, the highest depression scores were found in the range of 56 to 64 years, although they were within the normal range (Appendix A).

The mean self-perceived health-related quality of life score before vaccination was 7.5 ± 1.7, median 8 (IQR: 7–9), and after vaccination 7.4 ± 1.7, median 7 (IQR: 6–8.75). 

No significant differences were detected in the self-perceived quality of life of the patients between the different groups of renal therapy, both before vaccination and after vaccination. In both cases *p* = 0.12 (ANOVA test), but it is of note that kidney transplanted patients presented better results pre-vaccination (Appendix A). 

Likewise, no significant differences were detected in the patients’ self-perceived quality of life according to age groups, both before vaccination and after vaccination (*p*= 0.77 before vaccination and *p* = 0.40 after vaccination; ANOVA test; Appendix A).

### 3.2. Basal Descriptive Fears, Personal Coping Resources and Coping Resources Demanded of the Healthcare Team

This section explores patient fears and coping resources through three open-ended questions and two close-ended questions, namely: “Regarding the vaccine, what do you fear most?”, “Regarding vaccine-related fears, what do you think helps you feel better?”, and “How do you think we can help you from the hospital/hemodialysis clinic?”. No statistical differences were found among the different treatment modalities.

The results are detailed in Table 2, Table 3 and Table 4. 

## 4. Discussion

The present study yields some original and interesting results to further our understanding of the relationship between the COVID-19 vaccine and its psychological impact on the fears, resources, and health-related quality of life of 117 renal patients undergoing different treatment options. The main finding of the study is that the global mean of normal scores for anxiety and depression symptoms in chronic kidney disease patients measured by self-reported questionnaires (HADS-A and HADS-D) decreased post-vaccination. To our knowledge, this is the first study to report such findings for patients with CKD.

Although most patients presented scores that fell within the normal range on the anxiety subscale before vaccination, we found 8% of patients with scores above 11 points or suggestive of a clinical problem. After vaccination, the percentage decreased to only 1%. Similar results were obtained on the depression subscale, but in this case, the percentage of patients with results compatible with a clinical problem was very low (2% and 1%) both before and after vaccination.

Commonly, kidney patients present a decrease in their perception of their health and an increase in anxious−depressive states [16]. Some of the causes attributed to this are associated with comorbidity, lack of professional occupation, and complex therapeutic regimes [17]. The overall finding is that patients had lower scores in the second survey on both the anxiety and depression scales. In absolute terms, however, the mean remained within the normal range. It is fortunate that the vast majority of patients’ scores were located within this range. 

The World Health Organization has highlighted the importance of studying the consequences of the COVID-19 pandemic on the general population’s mental health. Some studies have found that elderly people demonstrated better psychological responses [18]. In our study, there were no statistically significant differences found in the anxiety or depression scores across the different age groups. Higher scores for anxiety were presented by patients in the youngest quartile (<58 age-range), while the highest depression scores were found in the 56–64 age range, but both scores are within the normal range and as such were not of clinical relevance. Fernández-Ballesteros and Sánchez-Izquierdo [19] conducted a survey study on an elderly Spanish population in a 60–93-year-old age group, and found emotional regulation and perceived control, high effectiveness, and stability of social interactions. It is clear that preventive health strategies, based on behavioral change, are a pending area of concern especially for older renal patients. Given that psychologists specialize in the area of behavioral change, it also seems clear that there is a significant role for them to play within renal interdisciplinary healthcare teams as activators of preventive programs. 

We did not find differences in anxiety or depression scores between modality groups, but there were interesting tendencies. Patients on hemodialysis barely changed their results in either anxiety or depression scores. It is of note that their scores were lower than those of other groups, especially before vaccination. Patients with advanced CKD not on dialysis presented the highest scores both for anxiety and depression. It is clear that more effort is needed in the promotion of wellbeing in those pre-dialysis patients [20].

No significant differences were found for depression, anxiety, and health-related quality of life (HRLQOL) scores among the different groups of patients before and after vaccination. It seems positive that patients on any renal replacement treatment program in our study scored their HRQOL between 7–8 points at the basal measure. Nevertheless, patients who had not yet started the program (advanced chronic kidney disease at pre-dialysis care) scored lower at 6.8–7. There is good evidence that the area of pre-dialysis education is an appropriate scenario in which to manage preventive psychological strategies regarding emotional states in order to assist patients in coping with uncertainty in the lead-up to complex and invasive dialysis treatments [21].

The three main fears reported by the participants at baseline were adverse effects, not getting the vaccine, and lack of information. This is in line with the hypothesis that the vaccine has a protective emotional effect against stress and the serious threats derived from the global situation of quarantine and the pandemic, as documented in other studies [22]. 

In addition, renal patients report that personalized, friendly, and close communication of updated information delivered by professional healthcare teams helped them to cope.

Our data, in addition to that from previous literature, support the importance of a renal multidisciplinary team, therapeutic communication, and emotional support strategies as ways to help patients deal with the stress derived from having a chronic health condition, by improving patient health outcomes and promoting wellbeing [22]. It is important to note that health objectives in complex chronic patients are not solely based on measuring biological parameters such as creatinine, glomerular filtration, and preserving body functions in general. They are also concerned with promoting the wellbeing of the patient, which encompasses both emotional and biological dimensions of a patient’s health.

Low vaccine acceptance among the general population is a serious threat to successful COVID-19 vaccination, as previous research has shown [23,24]. It is fortunate that among European countries, Spain has a high acceptance level of the vaccine. This variable, therefore, may have influenced kidney patient’s perception of the vaccine as an emotional resource rather than as a threat.

It must be noted that the size of the sample, recruited using convenience sampling and lacking a control group, is a limitation of this study. This is, however, the first study of its kind in Spain. Another limitation is the fact that not all the patients who started the study gave a second round of responses. This was due to the circumstances of the pandemic and the difficulty in making hospital visits. 

These limitations may be offset by some of the strengths of the study, such as the opportunity to test patients before receiving the SARS-CoV-2 vaccine, a situation that will no longer be easily repeated, as well as the inclusion of various modalities of chronic kidney disease. Despite all the obstacles, the study was successfully carried out during the time of the pandemic—a situation that made any kind of study difficult.

The psychological impact of the situation created by the appearance of new SARS-Cov-2 mutations, the need for a supplementary dose of vaccine, and the effect of all these factors on the prognosis of patients with CKD remains to be defined. New studies will be required to assess the impact of comprehensive vaccine coverage and further psychological impacts.

Nevertheless, these results are a stimulus for further consideration of the contrasted multidimensionality of the individual and of the value of teamwork. They also emphasize the enormous wealth generated by collaboration and cooperation during times of pandemic. In short, they reflect the benefits offered to renal patients of such an approach and the highly likely greater satisfaction it rewards to healthcare professionals.

## 5. Conclusions

We conducted a pre−post design study to assess the impact of vaccination against SARS-CoV-2 on anxiety and depression scores in patients with different modalities of chronic kidney disease in Spain. Vaccination improved psychological distress in renal patients with clinical scores. No significant differences were found in depression, anxiety, and health-related quality of life (HRLQOL) scores among the different groups of patients before and after vaccination. The findings of the present study could provide health practitioners with an empirical base from which to design a strategy that better promotes psychological wellbeing for future vaccination campaigns.

## Figures and Tables

**Figure 1 vaccines-10-00299-f001:**
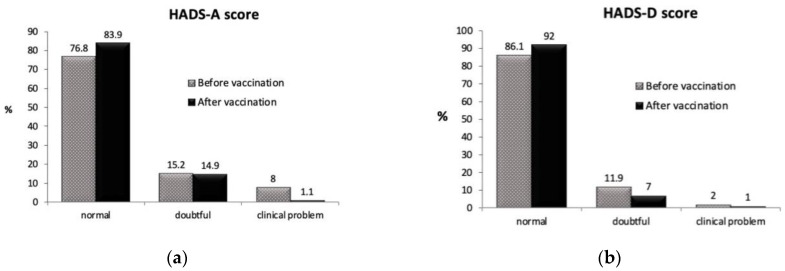
(**a**) Descriptive anxiety results in the HADS test in all participants before and after COVID-19 vaccination. Chi-square test *p* = 0.05. (**b**) Descriptive depression results in the HADS test in all participants before and after COVID-19 vaccination. Chi-square test *p* = 0.80. After vaccination: two weeks after the second dose of vaccination.

**Figure 2 vaccines-10-00299-f002:**
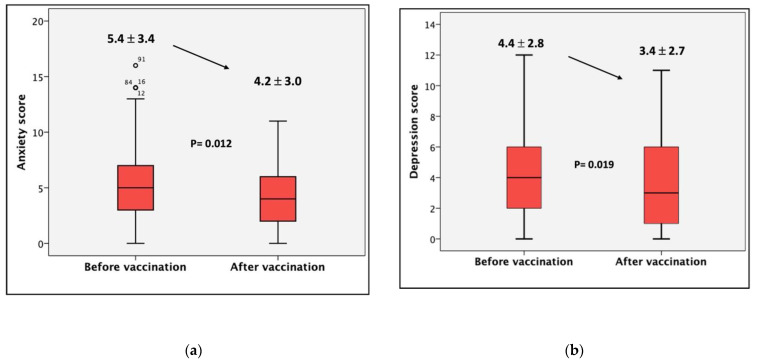
Comparative results of scores before and after vaccination (for all participants). The patients presented a significant improvement in both anxiety (*p* = 0.012) and depression (*p* = 0.019) scores two weeks post-vaccination. (**a**) The left panel shows anxiety score before vaccination and two weeks after the second dose of vaccination. Before vaccination: mean ± standard deviation: 5.4 ± 3.4, median: 5 (IQR: 3–7). After vaccination: mean standard deviation: 4.2 ± 3.0, median: 4 (IQR: 2–6). *p* = 0.012 Mann−Whitney U test. (**b**) The right panel shows the depression score before vaccination and two weeks after the second dose of vaccination. Before vaccination: mean ± standard deviation: 4.4 ± 2.8, median: 4 (IQR: 2–6). After vaccination: mean ± standard deviation: 3.4 ± 2.7, median: 3 (IQR: 1–5). *p* = 0.019 Mann−Whitney U test. Before vaccination, *n* = 117. After vaccination, *n* = 87.

**Figure 3 vaccines-10-00299-f003:**
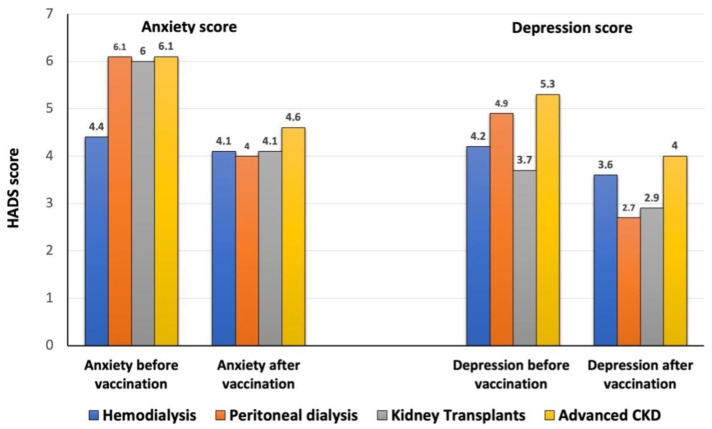
Mean anxiety and depression score by CKD groups. No significant differences were found between the groups. ANOVA test, *p* = 0.10.

**Table 1 vaccines-10-00299-t001:** Sociodemographic characteristics and clinical profile.

	HD	PD	KT	CKD 4/5	*p*
Age	70 (65–76.2)	71 (57.5–76)	60 (53–71.7)	65 (56.7–73)	0.32
Sex (male) *n* (%)	34 (68)	11 (84.6)	21 (65.6)	13 (59.1)	0.47
Previous COVID-19	4 (8)	1 (7.7)	2 (6.5)	0	0.69
Type 2 diabetes mellitus	20 (60)	5 (38.5)	7 (21.9)	8 (36.4)	0.38
Hypertension	42 (84)	11 (84.6)	27 (84.4)	17 (77.3)	0.89
Cardiopathy	18 (36)	6 (46)	3 (9.4)	3 (13.6)	0.08
Liver disease	4 (8)	0	0	0	0.13
Dialysis vintage	35 (14–66)	10 (2–20.5)	48.5 (25–84)	65 (56.7–73)	0.89
Kidney transplant vintage			29 (14–38.5)		

HD, hemodialysis; PD, peritoneal dialysis; KT, kidney transplantation; CKD 4/5, chronic kidney disease stage 4–5, not yet on dialysis.

**Table 2 vaccines-10-00299-t002:** Basal descriptive fears (N = 98) results in the total sample.

“Regarding Vaccine, What Do You Fear Most?”	*n* (%)
Nothing Adverse effects (i.e., thrombus)	44 (44.9) 34 (34.7)
Not having access to vaccine and get infected Lack of information	12 (12.2) 5 (5.1)
Vaccine brand	3 (3.1)

**Table 3 vaccines-10-00299-t003:** Basal descriptive personal coping resources (N = 101) results in the total sample.

“Regarding Vaccine Related Fears, What Do You Think Helps You Feel Better?”	*n* (%)
Family and friends The vaccine itself	47 (46.5) 15 (14.9)
Music, books and watching TV series Nature and walks	12 (11.9) 9 (8.9)
Trust in my healthcare team Physical activity No listening to the news Work	6 (5.9) 6 (5.9) 4 (4) 2 (2)

**Table 4 vaccines-10-00299-t004:** Basal descriptive coping resources demanded of the healthcare team (N = 106) results in the total sample.

“How Do You Think We Can Help You from the Hospital/Hemodialysis Clinic?”	*n* (%)
They are already doing it Personalized information, kindness, and close communication	53 (50) 31 (29.2)
Facilitating vaccine access Research	14 (13.2) 8 (7.5)

## Data Availability

The data underlying this article will be shared upon reasonable request to the corresponding author.

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
