# Peer review of "COVID-19 Vaccination Improved Psychological Distress (Anxiety and Depression Scores) in Chronic Kidney Disease Patients: A Prospective Study"

_vaccines, 2022, doi:10.3390/vaccines10020299_

Round 1

Reviewer 1 Report

It was a pleasure to read your work.

In the Introduction, you should show more new results from other researchers. I suggest to add the subsequent references:

  • Guerra F, Di Giacomo D, Ranieri J, Tunno M, Piscitani L, Ferri C. Chronic Kidney Disease and Its Relationship with Mental Health: Allostatic Load Perspective for Integrated Care. Journal of Personalized Medicine. 2021; 11(12):1367. https://doi.org/10.3390/jpm11121367
  • De Pasquale, C., Pistorio, M. L., Veroux, M., Indelicato, L., Biffa, G., Bennardi, N., Zoncheddu, P., Martinelli, V., Giaquinta, A., & Veroux, P. (2020). Psychological and Psychopathological Aspects of Kidney Transplantation: A Systematic Review. Frontiers in psychiatry11, 106. https://doi.org/10.3389/fpsyt.2020.00106
  • Iida, H., Fujimoto, S., Wakita, T., Yanagi, M., Suzuki, T., Koitabashi, K., Yazawa, M., Kawarazaki, H., Ishibashi, Y., Shibagaki, Y., & Kurita, N. (2020). Psychological Flexibility and Depression in Advanced CKD and Dialysis. Kidney medicine2(6), 684–691.e1. https://doi.org/10.1016/j.xkme.2020.07.004

In the methods section, it should be described also the exclusion criteria.

In Table 1 "DM" acronysm can be specified in the legend.

You should revise the English language and punctuation; several words are not correctly spelled and some sentences are difficult to understand.

In Data analysis section was not reported the level of significance adopted.

In the Method section you should specify if the assessments were conducted by online survey or in person. 

Author Response

REVIEWER #1

Dear reviewer, many thanks for the time that you expend on this manuscript. Please, see below the answers to your suggestions

Q1.- In the Introduction, you should show more new results from other researchers. I suggest to add the subsequent references:

  • Guerra F, Di Giacomo D, Ranieri J, Tunno M, Piscitani L, Ferri C. Chronic Kidney Disease and Its Relationship with Mental Health: Allostatic Load Perspective for Integrated Care. Journal of Personalized Medicine. 2021; 11(12):1367. https://doi.org/10.3390/jpm11121367
  • De Pasquale, C., Pistorio, M. L., Veroux, M., Indelicato, L., Biffa, G., Bennardi, N., Zoncheddu, P., Martinelli, V., Giaquinta, A., & Veroux, P. (2020). Psychological and Psychopathological Aspects of Kidney Transplantation: A Systematic Review. Frontiers in psychiatry11, 106. https://doi.org/10.3389/fpsyt.2020.00106
  • Iida, H., Fujimoto, S., Wakita, T., Yanagi, M., Suzuki, T., Koitabashi, K., Yazawa, M., Kawarazaki, H., Ishibashi, Y., Shibagaki, Y., & Kurita, N. (2020). Psychological Flexibility and Depression in Advanced CKD and Dialysis. Kidney medicine2(6), 684–691.e1. https://doi.org/10.1016/j.xkme.2020.07.004

A1.- We have included references to the very interesting suggested papers in the introduction.

In methods section we have described exclusion criteria as suggested. Without a doubt they have improved the initial approach of the article.

Q2.- In the methods section, it should be described also the exclusion criteria.

A2.- In methods section we have described exclusion criteria as suggested.

Q3.- In Table 1 "DM" acronyms can be specified in the legend.

A3.- In table 1 we have specified DM acronym in the legend.

Q4.- You should revise the English language and punctuation; several words are not correctly spelled and some sentences are difficult to understand.

A4.- A native English speaker specialized in scientific language has revised English language and punctuation.

Q5.- In data analysis section we have reported the level of significance adopted as required.

A5.- In Data analysis section was not reported the level of significance adopted.

Q6.-In the Method section you should specify if the assessments were conducted by online survey or in person. 

A6.- In the method section we have specified that the assessments were conducted in person.

Jose Luis Górriz, Helena Garcia-Llana and Nayara Panizo on behalf of the authors

Reviewer 2 Report

Dear Editor,

 the paper by Helena Garcia-Llana entitled: “  COVID-19 vaccination improved psychological distress (anxi- 2 ety and depression scores) in chronic kidney disease patients. A 3 prospective study.” Reported the impact of vaccination against SARS Cov 2 on  anxiety and depression scores in patients with different modalities of chronic kidney disease, a very peculiar setting of “immunocompromised” patients.

In the introduction setting is required:

- to add the date con the clinical outcome of COVID-19 in hemodialysis patients, peritoneal dialysis patients, kidney transplants and  advanced chronic kidney disease patients at pre-dialysis care.

- to add the date on the percent of acceptance vaccination anti-SARS-COV-2.

For me is able for publication after minor revision

Author Response

Dear reviewer, many thanks for the time that you expend on this manuscript. Please, see below the answers to your suggestions.

REVIEWER #2.

The paper by Helena Garcia-Llana entitled: “  COVID-19 vaccination improved psychological distress (anxiety and depression scores) in chronic kidney disease patients. A 3 prospective study.” Reported the impact of vaccination against SARS Cov 2 on  anxiety and depression scores in patients with different modalities of chronic kidney disease, a very peculiar setting of “immunocompromised” patients.

Q1.- In the introduction setting is required:

- to add the date con the clinical outcome of COVID-19 in hemodialysis patients, peritoneal dialysis patients, kidney transplants and  advanced chronic kidney disease patients at pre-dialysis care.

A1.- In the introduction we already mention that high mortality rates in CKD patients have been recorded after SARS Cov 2 infection, especially those patients with kidney transplants and on dialysis.

We have not analyzed clinical outcomes of COVID-19 in our study because it was out of our scope.

Q2.- - To add the date on the percent of acceptance vaccination anti-SARS-COV-2.

A2.- Most of participants in our study had not suffered from COVID-19 at baseline.

As currently specified in 2.1.3 Procedures, unwillingness to receive vaccination was an exclusion criterion, so 100 % of participants accepted vaccination. It is an excellent idea to clarify this fact in the manuscript and so we have done.

Jose Luis Górriz, Helena Garcia-Llana and Nayara Panizo on behalf of the authors